# Automated Quality Control of Cleaning Processes in Automotive Components Using Blob Analysis

**DOI:** 10.3390/s25092710

**Published:** 2025-04-24

**Authors:** Simone Mari, Giovanni Bucci, Fabrizio Ciancetta, Edoardo Fiorucci, Andrea Fioravanti

**Affiliations:** Dipartimento di Ingegneria Industriale e Dell’informazione e di Economia, Università dell’Aquila, 67100 L’Aquila, Italy; giovanni.bucci@univaq.it (G.B.); fabrizio.ciancetta@univaq.it (F.C.); edoardo.fiorucci@univaq.it (E.F.); andrea.fioravanti@univaq.it (A.F.)

**Keywords:** automotive manufacturing, blob analysis, computer vision, quality control, surface inspection

## Abstract

This study presents an automated computer vision system for assessing the cleanliness of plastic mirror caps used in the automotive industry after a washing process. These components are highly visible and require optimal surface conditions prior to painting, making the detection of residual contaminants critical for quality assurance. The system acquires high-resolution monochrome images under various lighting configurations, including natural light and infrared (IR) at 850 nm and 940 nm, with different angles of incidence. Four blob detection algorithms—adaptive thresholding, Laplacian of Gaussian (LoG), Difference of Gaussians (DoG), and Determinant of Hessian (DoH)—were implemented and evaluated based on their ability to detect surface impurities. Performance was assessed by comparing the total detected blob area before and after the cleaning process, providing a proxy for both sensitivity and false positive rate. Among the tested methods, adaptive thresholding under 30° natural light produced the best results, with a statistically significant z-score of +2.05 in the pre-wash phase and reduced false detections in post-wash conditions. The LoG and DoG methods were more prone to spurious detections, while DoH demonstrated intermediate performance but struggled with reflective surfaces. The proposed approach offers a cost-effective and scalable solution for real-time quality control in industrial environments, with the potential to improve process reliability and reduce waste due to surface defects.

## 1. Introduction

In modern industrial processes, ensuring the quality and reliability of components is essential to guarantee the proper functioning and longevity of the final products. High-tech industries such as the automotive, electronics, and aerospace sectors demand high-precision manufacturing to avoid defects, safety issues, and malfunctions [1]. In this context, automated vision systems have emerged as a key technology for monitoring production lines, performing quality control, and automating complex manufacturing tasks.

Computer vision systems are increasingly being adopted in industrial settings due to their ability to provide fast, accurate, and repeatable inspections of components. Unlike traditional human inspection methods, which are often subjective and prone to error, computer vision systems offer a consistent and objective evaluation of product quality [2]. The ability to process high-resolution images in real time and extract quantitative measurements has made computer vision an essential tool for detecting defects, verifying alignment, and optimizing manufacturing efficiency [3]. These systems have been successfully integrated into production lines to inspect surfaces, measure dimensions, and classify components. They are also used to monitor the correct positioning of objects, identify foreign particles, and control robotic systems in complex assembly operations [4].

A central technique in computer vision is blob analysis, which allows the identification and segmentation of connected regions in an image that share similar properties such as intensity, color, or texture. Blob analysis is particularly effective for detecting objects or defects with irregular shapes and varying sizes [5]. The process typically involves extracting pixel features, classifying connected regions, and segmenting individual blobs based on geometric and intensity criteria. Over the years, several algorithms have been developed to improve the accuracy and speed of blob analysis. Thresholding techniques, for example, convert the image into a binary map by setting an intensity threshold. The Laplacian of Gaussian (LoG) [6] method enhances blob-like structures by combining smoothing with edge detection, while the Difference of Gaussians (DoG) [7] technique approximates the LoG method by calculating the difference between two images filtered at different scales. The Determinant of Hessian (DoH) [8,9] method analyzes the Hessian matrix to identify blob-like regions with greater sensitivity to shape and scale. More recently, machine learning approaches have been introduced, allowing blob analysis systems to learn complex patterns and adapt to variations in shape and texture [10,11,12,13].

Blob analysis has found widespread application in both industrial and non-industrial fields. In industrial settings, it is used for surface inspection to detect cracks, scratches, and impurities on metal, plastic, and glass surfaces [14]. It also plays a key role in assembly verification, ensuring that all components are correctly positioned and properly assembled [15]. In object recognition, blob analysis enables the identification and sorting of products on conveyor belts and the alignment of parts in automated systems [16]. The technique is equally important in the medical field, where it is used for tumor detection, segmentation of cells in microscopic images, and identification of biomarkers in MRI scans [17,18,19]. In security and surveillance, blob analysis supports the detection of human presence or movement, while in remote sensing, it is employed for land classification and environmental monitoring using satellite images [20,21,22]. The versatility of blob analysis makes it an invaluable tool across multiple domains.

Despite the progress in blob analysis, several challenges remain when applying these techniques in industrial settings. One of the main issues is the variability in lighting conditions, which can produce reflections and shadows that obscure or distort the shapes of blobs [23]. The characteristics of the surface itself, such as transparency or reflectiveness, can also make it difficult to distinguish impurities from the background. The sizes and shapes of the objects being analyzed represent an additional complication: small or irregular particles may not be properly segmented using traditional methods. Furthermore, industrial systems often require real-time processing, which imposes strict constraints on computational complexity and algorithm execution time.

In this study, we address the problem of assessing the cleanliness of mirror caps used in the automotive industry after undergoing a cleaning process, using a computer vision system based on blob analysis. The goal is to develop an automated system capable of providing a quantitative evaluation of the surface cleanliness by detecting and segmenting small contaminants and impurities. To this end, images are acquired under different lighting conditions, both before and after the cleaning process. The acquired images are processed using a combination of blob analysis algorithms. The performance of these algorithms is compared to identify the most effective combination of lighting and processing technique for detecting impurities and assessing cleaning effectiveness. The results provide a basis for improving the reliability and efficiency of industrial quality control systems based on computer vision.

## 2. Materials and Methods

The goal of this work is to develop an automated system based on computer vision and blob analysis to assess the degree of cleanliness of an industrial component after undergoing a cleaning process. The system aims to provide a quantitative and objective evaluation of the cleaning effectiveness by detecting and segmenting contaminants or impurities on the surface of the component using different lighting and blob analysis techniques.

### 2.1. Experimental Setup

The study was conducted on plastic components used in the automotive industry, which presented different degrees of surface cleanliness. The tested samples included components in their initial state, components after undergoing a washing process, and components prepared for the painting phase. The goal was to detect the presence of dust particles or contaminants on the surface and assess the effectiveness of the cleaning process. To highlight the presence of contaminants, images were acquired under different lighting conditions. The lighting sources used in the study included natural light, infrared (IR) at 850 nm, and infrared at 940 nm. Each lighting condition was tested at multiple angles of incidence, specifically at perpendicular incidence, 30° inclination, 45° inclination, and grazing angle. Adjustments to light intensity and camera exposure were made to maintain consistent brightness and contrast across different setups. To ensure consistency in image acquisition, a fixed positioning system was employed. This system maintained a constant distance and orientation between the camera and the component under analysis, ensuring that the images were acquired under identical conditions for direct comparison.

A monochrome industrial camera (Teledyne DALSA Genie Nano-CL M2420, Teledyne DALSA, Waterloo, ON, Canada) was used for image acquisition. This device features a 2.36 MP CMOS sensor with global shutter, 5.5 μm pixel size, and Camera Link interface, ensuring high sensitivity, sharpness, and robustness against motion blur or reflections. The camera was mounted in a vertical orientation to acquire images perpendicularly to the component surface, which was placed flat on a reference table. Illumination was provided by configurable spotlights capable of delivering both natural and infrared light with variable beam widths. The lighting direction was manually adjusted to achieve the desired angle of incidence.

A schematic representation of the experimental setup is shown in Figure 1, which highlights the relative positioning of the test object, camera, and light source.

The decision to employ a monochrome camera instead of a color model was guided by the specific requirements of the application. In surface contamination analysis, the detection of subtle texture variations and low-contrast features is often more critical than the identification of chromatic cues. Monochrome sensors, by eliminating the color filter array, offer increased sensitivity to light and improved spatial resolution for a given sensor size. This enables more accurate detection of faint contaminants, such as dust or fine residues, especially under controlled lighting conditions. Moreover, color information was intentionally excluded to reduce the influence of ambient color temperature, surface pigmentation, and specular reflections, all of which could compromise the robustness and reproducibility of the analysis.

### 2.2. Image Acquisition and Preprocessing

A high-resolution monochrome industrial camera was used to acquire the images. Monochromatic imaging was chosen to eliminate color variations and focus on variations in texture and intensity. The acquisition environment was controlled to minimize the effect of ambient light and maintain stable imaging conditions. Figure 2 shows the images of clean components under different lighting configurations, highlighting the variations in intensity and texture visibility based on the lighting setup. Figure 3 shows the same components after contamination, providing a direct comparison of the effect of lighting and processing techniques on the visibility of contaminants. These figures illustrate, to the human eye, the influence of illumination wavelength, beam width, and incidence angle on defect detection.

To enhance the visibility of small contaminants and improve the performance of blob detection algorithms, a contrast enhancement technique based on Contrast-Limited Adaptive Histogram Equalization (CLAHE) [24] was applied to each image prior to analysis. CLAHE works by dividing the input image into small, non-overlapping regions, called tiles. A histogram of pixel intensities is computed for each tile. The histogram is then clipped at a predefined threshold to prevent over-amplification of noise or bright spots. The clipped values are redistributed across the histogram, and a transformation function is calculated to map the pixel intensities to an enhanced range. This process is repeated independently for each tile, allowing for localized contrast enhancement while preventing saturation in uniform regions.

Mathematically, the transformation function for each tile is defined as(1)T(x)=CDF(x) · (L−1),
where *T*(*x*) is the transformed pixel value, *x* is the original pixel value, *CDF*(*x*) is the cumulative distribution function of the histogram at intensity *x*, and *L* is the number of possible intensity levels (e.g., 256 for an 8-bit image).

To prevent over-enhancement of high-intensity regions, the histogram is clipped at a threshold value *T_c_*, defined as(2)$$T_{c}=\frac{\alpha \;\cdot \;N_{pixels}}{L},$$
where α is the contrast limit factor, $N_{pixels}$ is the number of pixels in the tile, and L is the number of intensity levels.

In this study, the contrast enhancement was configured by setting a contrast limit factor of 2, which defines the maximum allowable slope of the cumulative distribution function. The size of the local regions used for histogram calculation was set to 8 × 8 pixels, allowing localized contrast enhancement while maintaining a balance between local detail preservation and computational complexity.

### 2.3. Blob Analysis Algorithms

Several blob analysis algorithms were implemented and tested to evaluate their effectiveness in detecting surface contaminants under different lighting conditions. The goal was to identify the algorithm that maximizes detection accuracy while maintaining robustness to variations in illumination and surface texture. The performance of each algorithm was evaluated based on the total area of detected blobs, expressed in pixel^2^, which reflects the sensitivity of the algorithm to surface contaminants.

#### 2.3.1. Adaptive Thresholding

The adaptive thresholding algorithm segments the image by distinguishing pixels above and below a dynamically calculated intensity threshold. The image was first converted to grayscale and enhanced using CLAHE to improve local contrast. A Gaussian smoothing filter was subsequently applied to reduce high-frequency noise and enhance blob detection accuracy. The size of the Gaussian kernel was chosen to provide a balance between noise reduction and edge preservation. A binary threshold was then applied, setting pixel values above the threshold to white and those below to black.

The threshold for each pixel *T*(*x*,*y*) was computed based on the local mean intensity over a window of size w×w centered at that pixel, defined as(3)$$T(x,y)=\frac{1}{w^{2}}\sum_{i=-w/2}^{w/2}\sum_{j=-w/2}^{w/2}I(x+i,y+j),$$
indicating with I(x,y) the pixel intensity at location (x,y).

After thresholding, connected components were extracted, and small artifacts were removed by discarding objects below a predefined minimum area threshold. The total area of the detected blobs was computed by summing the areas of the remaining contours. An example of the results obtained using the adaptive thresholding algorithm under grazing natural light conditions is shown in Figure 4.

The adaptive threshold was dynamically adjusted based on local mean intensity and standard deviation, as defined in Equation (3), to account for different pollutant types and surface textures. A minimum area filter was applied to eliminate isolated noise, while the Gaussian kernel was empirically selected (size = 5 × 5) to preserve the visibility of small but significant contaminants. The parameters were tuned based on visual inspection of multiple pre-wash images to balance noise rejection and defect detection.

#### 2.3.2. Laplacian of Gaussian

The LoG algorithm [25] enhances blob-like structures by combining a smoothing step with edge detection. The method relies on the Laplacian operator, which measures the second-order intensity variations of the image, highlighting regions of rapid intensity change that typically correspond to edges of blob-like structures. The Laplacian operator is defined as(4)$$\nabla^{2}L\left(x,y\right)=\frac{\partial^{2}L}{\partial x^{2}}+\frac{\partial^{2}L}{\partial y^{2}},$$
where $L\left(x,y\right)$ is the smoothed image at position (x,y). The Laplacian measures the sum of the second-order derivatives of the image, which corresponds to the curvature of the intensity surface. Zero-crossings of the Laplacian response indicate potential blob edges.

To reduce noise and small-scale texture variations, the input image was first smoothed using a Gaussian filter with a kernel size of 5 × 5 pixels and a standard deviation of 0. After smoothing, the Laplacian operator was applied to the filtered image using a kernel size of 9 × 9 pixels to enhance regions of rapid intensity change.

A thresholding operation was then performed to isolate significant blob regions. The threshold was dynamically calculated based on the mean and standard deviation of the Laplacian response, according to the following relationship:(5)T=µ+k·σ,
where μ represents the mean intensity of the Laplacian response, σ is the corresponding standard deviation, and k is the thresholding factor, set at 0.15.(6)$$\nabla^{2}L\left(x,y\right)=\frac{\partial^{2}L}{\partial x^{2}}+\frac{\partial^{2}L}{\partial y^{2}},$$

This approach adapts the threshold to local contrast variations in the image, improving the algorithm’s sensitivity to subtle intensity variations.

After thresholding, a morphological opening operation using a 3 × 3 structuring element was applied, to remove small artifacts and noise. This method is particularly effective at detecting blobs with circular or elliptical shapes, although its sensitivity depends on the size of the Gaussian and Laplacian kernels, which need to be tuned to the scale of the expected blobs. An example of the results obtained using the LoG algorithm under grazing natural light conditions is shown in Figure 5.

#### 2.3.3. Difference of Gaussians

The DoG algorithm approximates the LoG method by subtracting two Gaussian-filtered versions of the same image, obtained with different kernel sizes. The image was first smoothed using a Gaussian filter with a larger kernel ($\sigma_{1}$) to reduce noise and enhance large-scale structures. A second Gaussian filter with a smaller kernel ($\sigma_{2}$) was then applied to preserve fine details. The difference between these two smoothed images was computed as(7)$$DoG\left(x,y\right)=G_{\sigma_{1}}\left(x,y\right)-G_{\sigma_{2}}\left(x,y\right),$$
where $G_{\sigma }\left(x,y\right)$ is the Gaussian-filtered image with standard deviation σ, computed over a kernel of size of k×k.

The resulting image was then thresholded using an adaptive threshold defined as in (5). For DoG, the thresholding factor was set to 0.2. The binary thresholded image was processed to extract the contours of the detected blobs. A morphological opening operation was applied using a structuring element of the size 3 × 3 pixels to remove noise and small artifacts. Only connected components with an area exceeding a predefined minimum threshold were retained, and their total area was computed. In this study, the values for the size of the Gaussian kernel were set to 5 × 5 for large-scale smoothing and 9 × 9 for finer details.

This method is computationally more efficient than LoG and can detect blobs of varying sizes, but its sensitivity depends on the choice of kernel sizes and the thresholding factor. An example of the results obtained using the DoG algorithm under grazing natural light conditions is shown in Figure 6.

#### 2.3.4. Determinant of Hessian

The DoH algorithm [26] detects blobs based on second-order intensity variations. The image was smoothed using three Gaussian filters with different kernel sizes to capture blob-like structures at different scales. The values used for the Gaussian kernel sizes were 5 × 5, 7 × 7, and 9 × 9. The Hessian matrix H(x,y) at each pixel location was computed as(8)$$H\left(x,y\right)=\left[\begin{array}{cc} \frac{\partial^{2}L}{\partial x^{2}} & \frac{\partial^{2}L}{\partial x\partial y}\\ \frac{\partial^{2}L}{\partial y\partial x} & \frac{\partial^{2}L}{\partial y^{2}} \end{array}\right],$$
where L(x,y) is the smoothed image at pixel location (x,y). The determinant of the Hessian was then computed as(9)$$\det\left(H\right)=\frac{\partial^{2}L}{\partial x^{2}}\cdot \frac{\partial^{2}L}{\partial y^{2}}-{\left(\frac{\partial^{2}L}{\partial x\partial y}\right)}^{2},$$

A threshold was applied to the determining response using the adaptive threshold defined in (5), with a thresholding factor of 0.1. Following the thresholding step, a morphological opening operation using a structuring element of the size 3 × 3 pixels was applied to remove small artifacts and noise. This method is particularly effective at detecting blobs with varying sizes and shapes, even in the presence of background texture, but it is computationally more demanding than other methods due to the multi-scale processing. An example of the results obtained using the DoH algorithm under grazing natural light conditions is shown in Figure 7.

## 3. Evaluation Metrics

The performance of the blob detection algorithms was evaluated based on the total detected blob area, expressed in pixels. This metric provides a direct measure of the extent of surface contaminants detected and reflects the sensitivity of the algorithm to variations in texture and lighting conditions. Conventional evaluation metrics such as precision, recall, and F1-score were not applicable in this study due to the absence of a pixel-level ground truth or manually annotated defect masks. In the context of industrial quality control, generating accurate annotations for surface contaminants—especially those varying in shape, texture, and reflectivity—is extremely time-consuming and often impractical. As a result, the total blob area was adopted as a proxy for detection sensitivity, providing an objective and scalable indicator of the algorithm’s ability to respond to surface irregularities under varying lighting conditions. Larger blob areas indicate a higher detection rate and improved defect localization capability. However, an excessive number of detected blobs may also reflect the presence of false positives or image noise, potentially affecting the overall precision of the algorithm. For this reason, the blob area was used as a proxy for detection sensitivity, with the awareness that it does not directly quantify classification accuracy, but rather the algorithm’s ability to respond to visible irregularities under varying lighting conditions. The comparison was conducted on both clean and contaminated components to evaluate the robustness of each algorithm to variations in illumination and surface texture. The performance of the algorithms was analyzed under natural and infrared (IR) lighting at wavelengths of 850 nm and 940 nm, considering variations in beam width (wide vs. narrow) and angle of incidence. This setup reflects real-world industrial inspection scenarios, where the primary objective is to maximize contamination detection without relying on extensive ground truth labeling. The evaluation aimed to identify the most effective combination of algorithm and lighting configuration for accurate and consistent detection of surface contaminants.

## 4. Experimental Results

The goal of this study was to develop an automated system capable of evaluating the quality of the cleaning process of industrial components. The system was designed to identify and segment contaminants remaining on the surface of the components after the washing process, using different blob detection algorithms and lighting configurations. The results presented in this section provide a comparative evaluation of the performance of the implemented algorithms under different lighting conditions, both before and after the washing process. The analysis focuses on identifying the most effective combination of algorithm and lighting configuration for accurate and consistent defect detection.

In the pre-wash phase, the adaptive thresholding method detected the largest blob areas, ranging from 139,729 pixel^2^ under grazing infrared light at 940 nm (wide spot) to 1,030,940 pixel^2^ under 30° natural light (wide spot), as shown in Table 1. This suggests that the thresholding method is more sensitive to surface irregularities, especially under natural light configurations. The LoG method exhibited high sensitivity, with blob areas ranging from 377,383 pixel^2^ to 978,897 pixel^2^, with the highest value observed under 45° natural light. The DoG method reported blob areas between 248,255 pixel^2^ and 888,653 pixel^2^, while the DoH method produced blob areas between 182,175 pixel^2^ and 934,481 pixel^2^, showing balanced performance under different lighting configurations. The highest sensitivity for both the DoG and DoH methods was recorded under 45° natural light, suggesting that this configuration enhances the detection of intensity variations associated with surface impurities.

In the post-wash phase, the total detected blob area decreased significantly across all methods, confirming the removal of surface contaminants after the washing process, as shown in Table 2.

The adaptive thresholding method again reported the highest blob areas, ranging from 6343 pixel^2^ under grazing infrared light at 940 nm (narrow spot) to 59,285 pixel^2^ under 45° natural light, suggesting that it maintains superior sensitivity even in the presence of lower contamination levels. The LoG method detected blob areas between 61,210 pixel^2^ and 150,705 pixel^2^, while the DoG method reported values between 12,388 pixel^2^ and 94,322 pixel^2^. The DoH method showed the lowest response, with detected blob areas ranging from 13,309 pixel^2^ to 71,974 pixel^2^, indicating reduced sensitivity to smaller impurities in the post-wash phase.

To highlight the impact of illumination type on algorithm performance, a comparative analysis was carried out between natural light and infrared (IR) configurations. Across all methods, natural light—especially at 30° and 45° incidence—consistently yielded higher blob areas, indicating improved sensitivity to surface defects. This behavior is attributed to the broader spectrum and diffuse reflections typical of natural light, which enhance texture visibility. In contrast, IR illumination at 850 nm and 940 nm exhibited more stable performance in terms of repeatability but reduced sensitivity, especially under grazing conditions, due to lower surface contrast.

Figure 8, Figure 9, Figure 10 and Figure 11 illustrate the blob detection results using the four different algorithms applied to the test samples in pre-wash and post-wash conditions. Each figure compares the blob area detected in pre-wash and post-wash conditions, providing insight into the sensitivity and performance of each algorithm in different lighting and spot size configurations.

Figure 12, Figure 13, Figure 14 and Figure 15 show examples of blob detection results obtained under the lighting conditions where each algorithm demonstrated the highest capability to quantify surface contamination. Figure 11 illustrates the outcome of adaptive thresholding under 30° natural light conditions, while Figure 12, Figure 13 and Figure 14 display the results of LoG, DoG, and DoH methods, respectively, under the lighting settings that yielded the most consistent blob detection.

Among the tested algorithms, adaptive thresholding exhibited the best overall performance in detecting surface contamination. This method was able to detect a larger contaminated area with greater consistency and fewer false positives. In contrast, LoG and DoG highlighted smaller contaminated areas and were more prone to false positives when applied to the clean surface, often identifying non-contaminant elements as blobs. The DoH algorithm showed moderate performance compared to LoG and DoG, as it captured a larger contaminated area and was less sensitive to noise. However, the DoH method struggled to handle reflections from the surface, which led to inaccuracies in blob detection under certain lighting conditions.

Across all pre-wash configurations, the setup combining adaptive thresholding with 30° natural light produced a statistically significant peak in detection performance, with a z-score of +2.05. This value is more than two standard deviations above the mean, making it the most sensitive configuration for surface contamination detection before washing.

In addition to evaluating detection sensitivity, a comparative analysis of the computational performance of each blob detection algorithm was carried out. Execution times were measured by isolating the processing phase from any input/output operations.

Table 3 presents the average execution times and standard deviations for the four considered methods: adaptive thresholding, LoG, DoG, and DoH. Among them, adaptive thresholding exhibited the highest computational cost, with an average of 485.08 ms per image. This is primarily attributed to its per-pixel adaptive thresholding mechanism, which requires localized calculations across the entire image. The LoG and DoH methods also showed significant processing times due to the application of large-kernel convolutions and second-order derivative operators. The DoG method, by contrast, achieved the lowest average execution time (99.97 ms), benefiting from its use of two Gaussian filters and a simple image difference operation.

## 5. Discussion

One of the main challenges observed during the evaluation of blob-based detection methods was the presence of false positives, especially in conditions involving reflective surfaces, lighting gradients, or complex textures. Gradient-based methods such as LoG and DoG showed a higher tendency to detect non-contaminant features, including edges, shadows, or surface curvature transitions, as potential defects. This issue was particularly evident under grazing light, where high contrast at material discontinuities amplified spurious detections. Conversely, adaptive thresholding exhibited greater robustness on clean surfaces, likely due to its locally adaptive nature, which reduces sensitivity to large-scale gradients. Nonetheless, even this method occasionally produced false detections due to minor surface marks or uneven illumination. Morphological filtering was used to reduce small or isolated regions, but some false positives persisted.

The optimal balance between sensitivity and accuracy was defined based on a dual criterion: a high blob area in the pre-wash phase (indicating effective contaminant detection) and a low blob area in the post-wash phase (minimizing false positives). Among the tested methods, adaptive thresholding combined with 30° natural light produced the highest z-score in the pre-wash phase while maintaining relatively low blob detection in the post-wash phase, demonstrating both sensitivity and specificity. This dual behavior is considered indicative of an optimal trade-off for industrial inspection.

## 6. Conclusions

This study presented an automated quality control system for assessing the cleanliness of automotive components after a washing process, based on blob analysis techniques. The system was designed to provide a quantitative evaluation of surface contamination by detecting and segmenting blobs corresponding to impurities or residues. Four blob analysis methods were evaluated under various lighting configurations, including adaptive thresholding, LoG, DoG, and DoH. The performance of the algorithms was compared based on the total detected blob area before and after the washing process, providing insights into the sensitivity and robustness of each method.

Among the tested algorithms, adaptive thresholding demonstrated the highest sensitivity and accuracy in detecting surface contamination, particularly under natural light at 30° incidence. This method consistently identified larger contaminated areas and produced fewer false positives when applied to clean surfaces. Its responsiveness to surface irregularities and robustness under varying lighting conditions make it the most reliable approach for industrial surface cleanliness assessment. The LoG and DoG methods showed lower sensitivity and were more prone to false positives, especially under certain lighting conditions where edges or texture gradients were misclassified as contaminants. The DoH algorithm offered intermediate performance but was affected by reflections and surface texture variability.

The experimental results revealed several challenges for practical deployment, including false positives due to specular reflections, the sensitivity of gradient-based methods to non-contaminant features, and the relatively high computational cost of some algorithms. These limitations directly motivate future work. Improving algorithmic robustness will aim to suppress spurious detections in the presence of reflections and lighting gradients, while optimizing execution time will enable real-time integration in production environments. Moreover, the adoption of hybrid approaches that combine traditional blob detection with machine learning could further enhance detection specificity and reduce false positives in complex inspection scenarios.

## Figures and Tables

**Figure 1 sensors-25-02710-f001:**
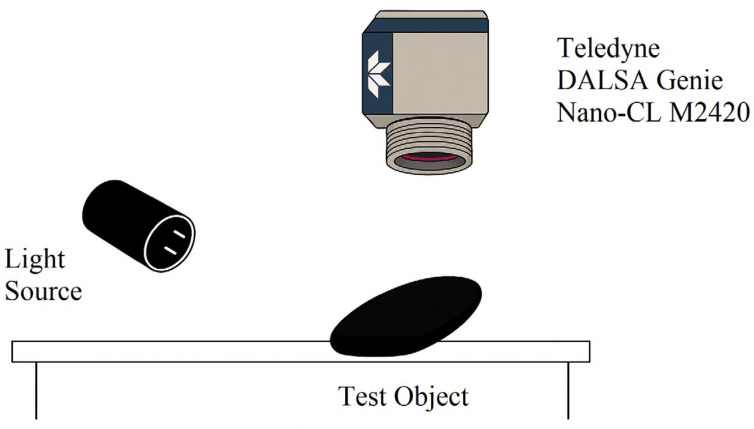
Schematic representation of the experimental imaging setup used for surface contamination analysis.

**Figure 2 sensors-25-02710-f002:**
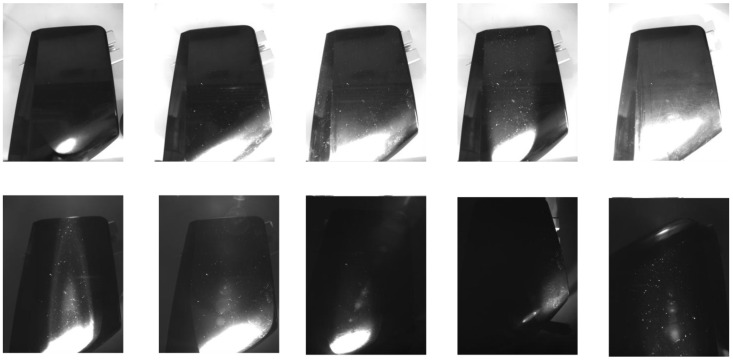
Comparison of different lighting configurations for detecting contaminants on clean industrial components. The images in the first row show the component under different natural light conditions, with variations in the angle of incidence and beam width. The images in the second row show the same component under different configurations of infrared illumination (850 nm and 940 nm), highlighting the effect of wavelength and angle of incidence on the visibility of surface defects.

**Figure 3 sensors-25-02710-f003:**
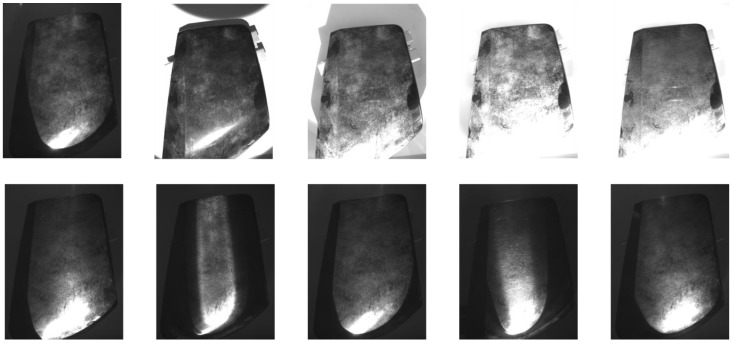
Comparison of different lighting configurations for detecting contaminants on dirty industrial components. The images in the first row show the component under different natural light conditions, with variations in the angle of incidence and beam width. The images in the second row show the same component under different configurations of infrared illumination (850 nm and 940 nm), highlighting the effect of wavelength and angle of incidence on the visibility of surface contaminants.

**Figure 4 sensors-25-02710-f004:**
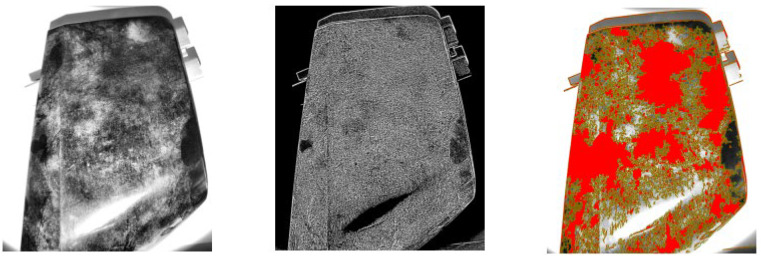
Example of adaptive thresholding under grazing natural light conditions. Original image after contrast enhancement using CLAHE (**left**). Output of adaptive thresholding highlighting potential defects and contaminants (**center**). Final blob detection result, with detected blobs overlaid in red (**right**).

**Figure 5 sensors-25-02710-f005:**
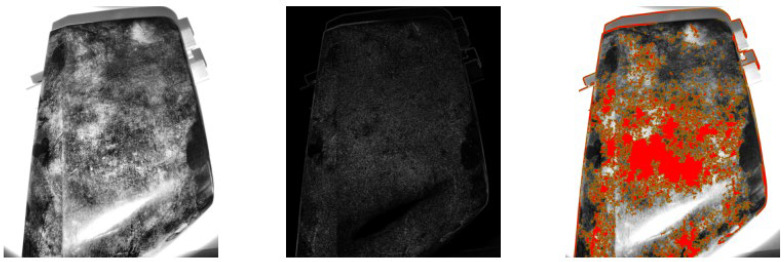
Example of LoG under grazing natural light conditions. Original image after contrast enhancement using CLAHE (**left**). Output of the Laplacian operator highlighting intensity changes and potential blob edges (**center**). Final blob detection result, with detected blobs overlaid in red (**right**).

**Figure 6 sensors-25-02710-f006:**
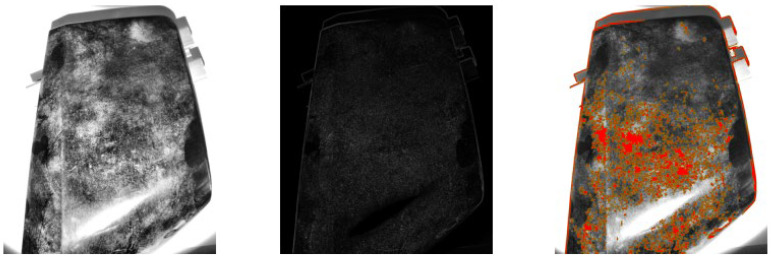
Example of DoG under grazing natural light conditions. Original image after contrast enhancement using CLAHE (**left**). Output of the DoG operator emphasizing regions of intensity difference at different scales (**center**). Final blob detection result, with detected blobs overlaid in red (**right**).

**Figure 7 sensors-25-02710-f007:**
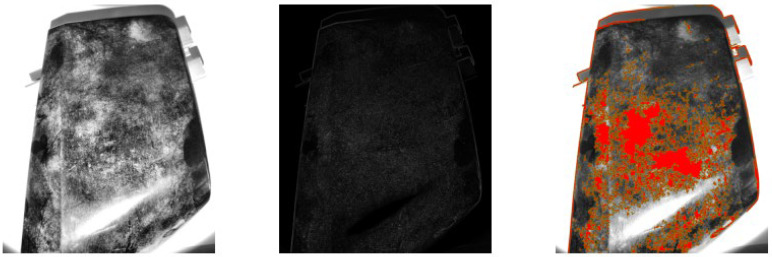
Example of DoH under grazing natural light conditions. Original image after contrast enhancement using CLAHE (**left**). Output of the Hessian matrix determinant highlighting regions of high second-order intensity variation (**center**). Final blob detection result, with detected blobs overlaid in red (**right**).

**Figure 8 sensors-25-02710-f008:**
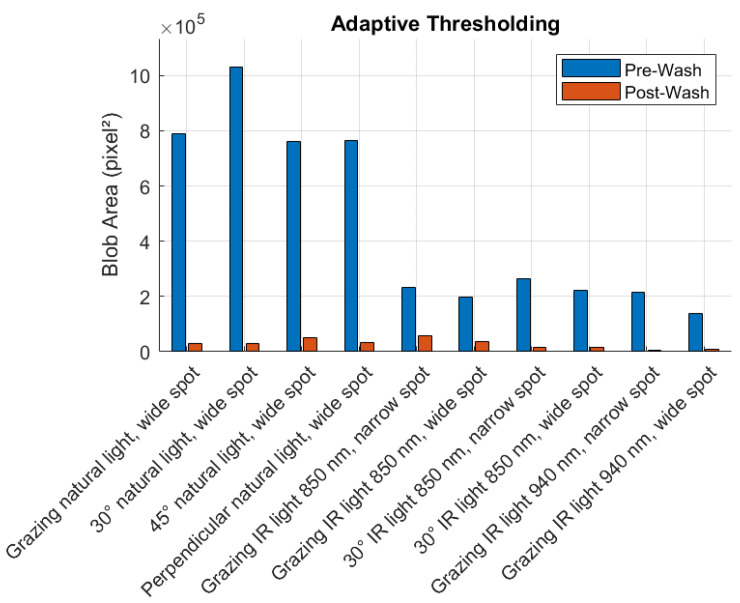
Pre-wash and post-wash blob detection results using adaptive thresholding.

**Figure 9 sensors-25-02710-f009:**
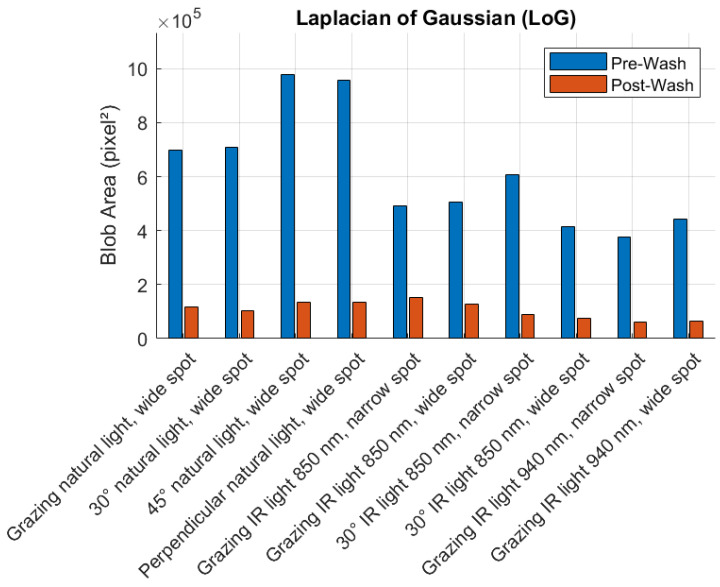
Pre-wash and post-wash blob detection results using the LoG algorithm.

**Figure 10 sensors-25-02710-f010:**
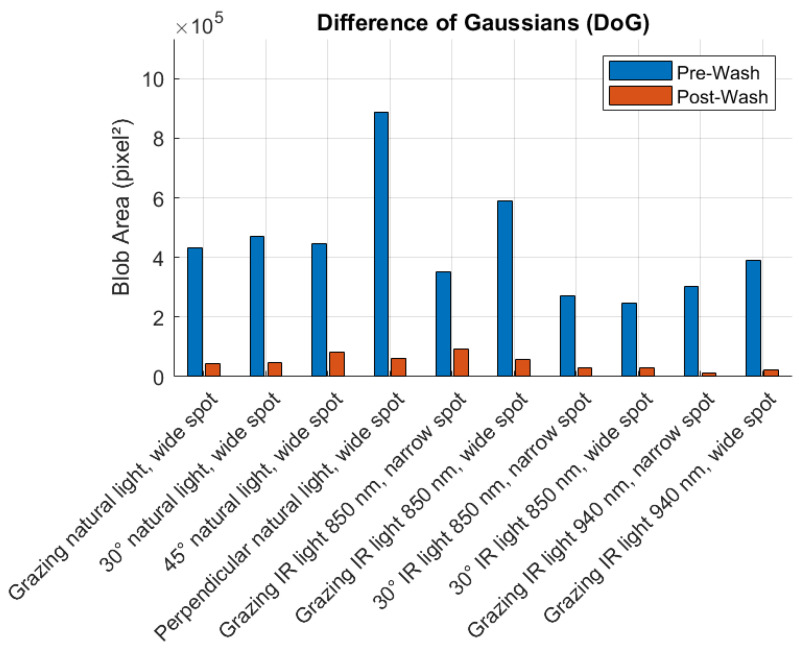
Pre-wash and post-wash blob detection results using the DoG algorithm.

**Figure 11 sensors-25-02710-f011:**
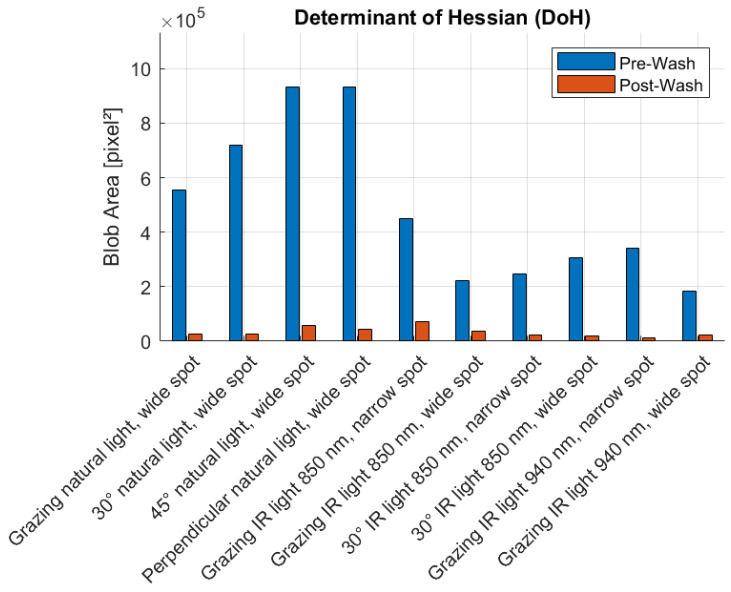
Pre-wash and post-wash blob detection results using the DoH algorithm.

**Figure 12 sensors-25-02710-f012:**
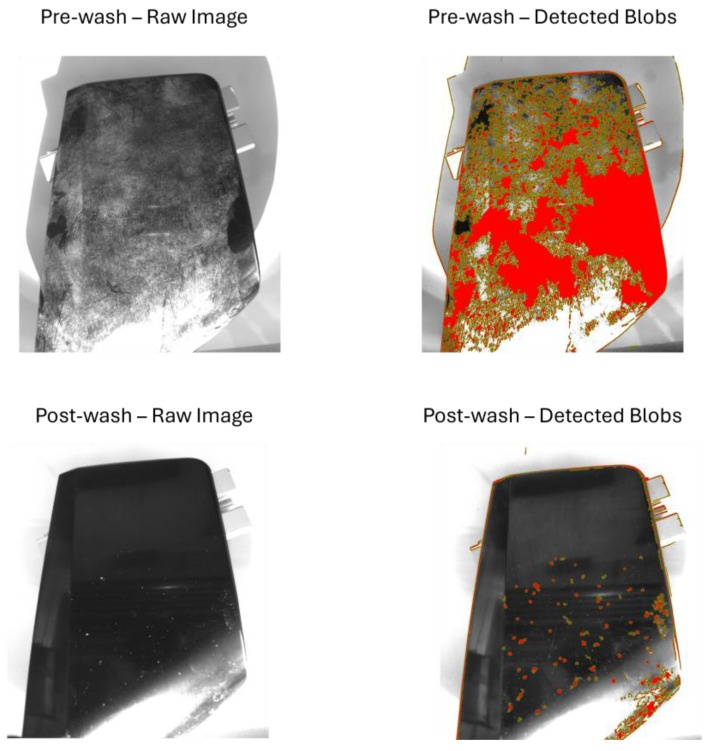
Example of adaptive thresholding under 30° natural light conditions.

**Figure 13 sensors-25-02710-f013:**
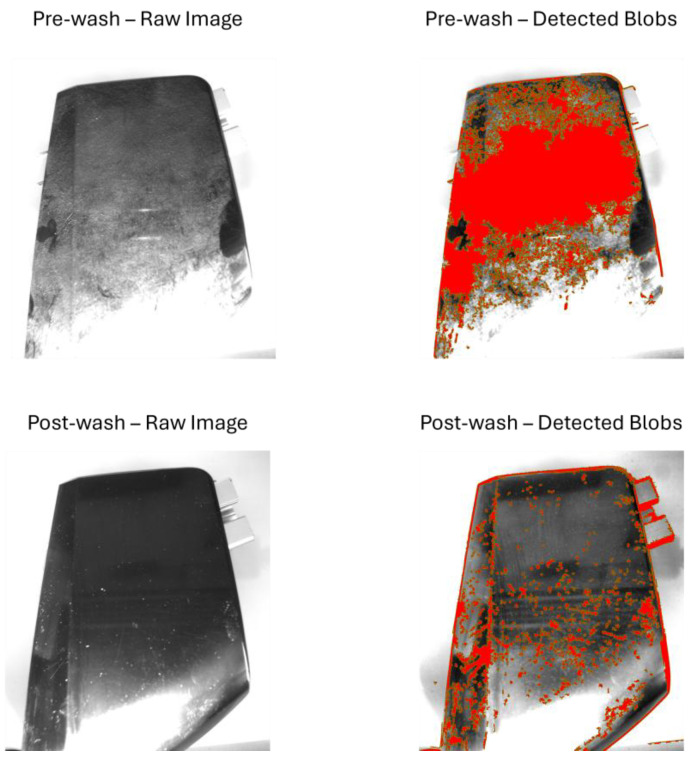
Example of LoG under 45° natural light conditions.

**Figure 14 sensors-25-02710-f014:**
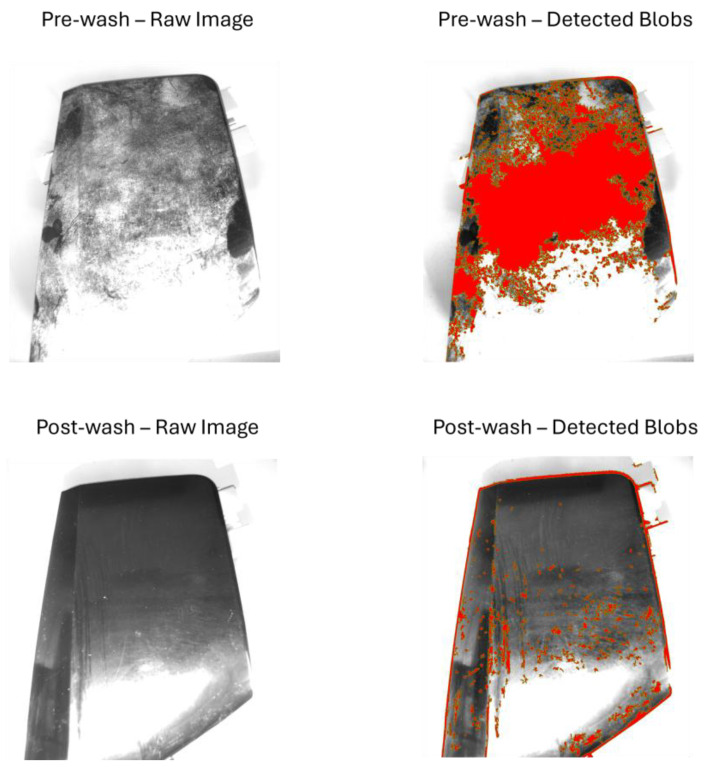
Example of DoG under perpendicular natural light conditions.

**Figure 15 sensors-25-02710-f015:**
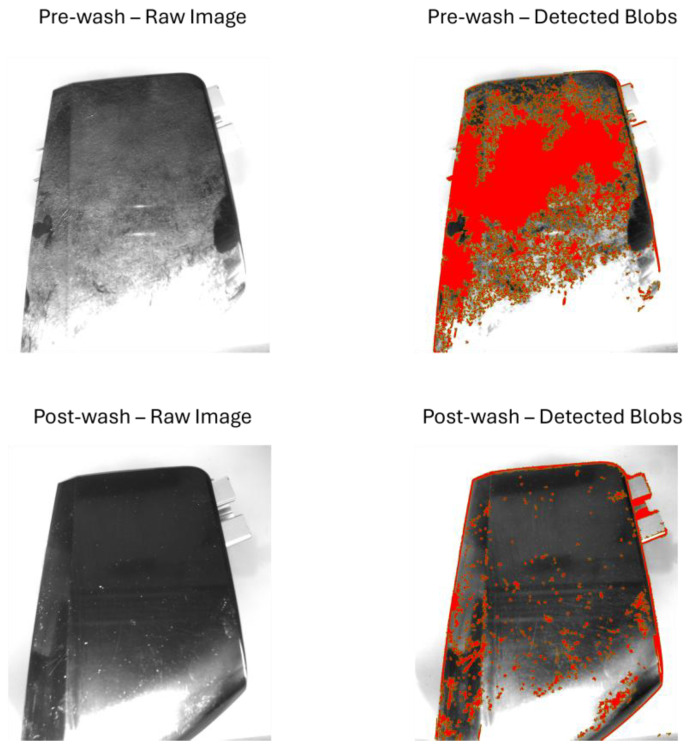
Example of DoH under 45° natural light conditions.

**Table 1 sensors-25-02710-t001:** Results of image processing in the pre-wash phase.

Test Conditions	Thresholding [pixel^2^]	LoG [pixel^2^]	DoG [pixel^2^]	DoH [pixel^2^]
Grazing natural light, wide spot	789,258	698,417	431,917	555,308
30° natural light, wide spot	1,030,940	708,567	470,773	720,016
45° natural light, wide spot	760,870	978,897	445,268	934,481
Perpendicular natural light, wide spot	763,908	955,641	888,653	932,220
Grazing IR light 850 nm, narrow spot	234,407	491,871	351,097	448,444
Grazing IR light 850 nm, wide spot	199,204	507,366	591,115	223,463
30° IR light 850 nm, narrow spot	265,645	606,043	269,963	246,913
30° IR light 850 nm, wide spot	221,776	416,314	248,255	304,662
Grazing IR light 940 nm, narrow spot	214,405	377,383	303,756	340,934
Grazing IR light 940 nm, wide spot	139,729	443,609	388,858	182,175

**Table 2 sensors-25-02710-t002:** Results of image processing in the post-wash phase.

Test Conditions	Thresholding [pixel^2^]	LoG [pixel^2^]	DoG [pixel^2^]	DoH [pixel^2^]
Grazing natural light, wide spot	28,801	117,494	44,773	26,461
30° natural light, wide spot	29,515	102,329	46,008	27,166
45° natural light, wide spot	50,564	134,073	83,671	59,051
Perpendicular natural light, wide spot	34,286	135,205	62,824	44,104
Grazing IR light 850 nm, narrow spot	59,285	150,705	94,322	71,974
Grazing IR light 850 nm, wide spot	35,397	129,456	58,910	35,236
30° IR light 850 nm, narrow spot	15,704	87,772	28,656	21,261
30° IR light 850 nm, wide spot	14,406	76,578	29,717	20,754
Grazing IR light 940 nm, narrow spot	6343	61,210	12,388	13,309
Grazing IR light 940 nm, wide spot	10,215	64,144	21,292	22,481

**Table 3 sensors-25-02710-t003:** Computational performance comparison of blob detection algorithms.

	Thresholding [ms]	LoG [ms]	DoG [ms]	DoH [ms]
**Execution time**	**485.08**	392.21	99.97	309.11

## Data Availability

The data presented in this study are not publicly available due to privacy restrictions.

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
