# Peer review of "Automated Quality Control of Cleaning Processes in Automotive Components Using Blob Analysis"

_sensors, 2025, doi:10.3390/s25092710_

Round 1
Reviewer 1 Report
Comments and Suggestions for Authors
The manuscript presents an automated inspection system leveraging image-based analysis to assess the cleanliness of automotive components post-washing. The authors compared four blob detection algorithms, adaptive thresholding, Laplacian of Gaussian (LoG), Difference of Gaussians (DoG), and Determinant of Hessian (DoH), under varied lighting conditions (natural and infrared) and some predefined angles. They evaluated these algorithms based on the total detected blob area as a measure of sensitivity to contaminants. A valuable benchmark for their performance in detecting surface contaminants was provided. As a result, they identified optimal method for contaminant detection - adaptive thresholding, particularly under natural light at a 30° incidence angle, outperformed the other methods in terms of sensitivity and consistency, though lighting variability and reflective surfaces impacted the method performance. As future work, the authors plan to improve influence of surface reflections, provide real-time processing capabilities to satisfy industrial needs, and to integrate alternative approaches, such as machine learning-based methods.
Overall, the manuscript is well structured and clearly written, and provides a practical contribution to automated quality control in automotive manufacturing. However, there are comments and suggestions to improve.
- The manuscript mentions that adaptive thresholding demonstarated the highest sensitivity and accuracy but doesn't discuss computational efficiency of all methods. No mention about execution time per method and their comparison. This might be critical for real-time processing capability which the authors plan to consider in future.
- The authors suggest to use total detected blob area as the evaluation metric but they did not explain why this metric was chosen and if it is the best one. Maybe other metrics, as standard in defect detection, precision, recall, F1-score, or some other, could provide a more comprehensive evaluation.
- The manuscript uses grayscale images to eliminate color variations, not explaining why monochrome camera was chosen, in spite of some contaminants maybe color-specific. Excluding color information might reduce detection accuracy for certain types of contaminants (oil stains, grease, rust or old paint residues).
- More detailed investigation and discussion of false positive (not just the mentioned edge artifacts) is desirable. Some ways of solution of this problem will be useful and at least can be planned for the future.
- The authors conclude that the adaptive thresholding method has the advantage basing on the study of a single sample. Does the shape or size of the sample matter? To be convincing the results should be replicated across multiple samples of different shape and size.
- Differences in blob areas between algorithms are presented descriptively to compare them visually only. Some mathematics (statistical tests or so) would be more convinсing here. Especially in the case the adaptive thresholding and DoH method results which are close to each other.
- In 2.1 and 2.2 sections the authors mention “fixed positioning system” and “a high-resolution monochrome industrial camera” which were employed. It would be desirable to provide more detailed information about the equipment used, as well as a photo of the setup used for image acquisition.
- Lines 306-312: When describing specific values for min and max ranges of largest blob areas, the authors often specify them in the text incorrectly, for example, for LoG method it is specified from 698417 (min) to 978897 (max), although the min value is 377383 (see Table 1). The same is for DoG and DoH cases, and also for Table 2 text description (lines 325-330), cases for Thresholding and LoG methods.
- Lines 50-51: Authors write: …”More recently, machine learning approaches have been introduced, allowing blob analysis systems to learn complex patterns and adapt to variations in shape and texture.” References to sources of information are desirable here. Recent hybrid models based on CNNs or transformers might outperform traditional methods in defect detection.
Reviewer 2 Report
Comments and Suggestions for Authors
1,Introduction
The Blob is introduced from the practical needs, then the history of the BIob is introduced, and its wide range of applications is shown. Finally, the challenges are presented: (1) shadows created by the illumination blur the shape of the blob (2) real-time is not enough (3) it is not possible to correctly segment the small particles and irregular particles. Finally, we introduce our own visualization system: it is able to detect and segment the fine contaminants, and to have a quantitative assessment of the cleanliness. In this paper, we compare and combine Blob analysis to find out the optimal lighting and technology for detecting contaminants.
Insufficient innovation, only in the combination of the original algorithm and different lighting experiments to compare with each other, research is weak, there is a certain value of factory production applications, providing a combination of car parts cleaning method. From the back to look at doing experimental work, there is a certain amount of workload.
2,Materials and Methods
More experiments are done, as well as relevant algorithmic substitution analysis and mathematical formula modeling in conjunction with their own projects, but relevant experimental details are missing to enhance the authenticity of the experiments.
Suggestion: add the theoretical basis for the selection of certain parameters, otherwise the paper lacks rigor:
Why is the choice of contrast limiting factor α shown in lines 154-158 a 2? What if this 2 affects the fairness of the comparison of the four algorithms in your BIOB analysis? Then you are drawing conclusions that have not been subjected to rigorous experimentation.
Suggestion: add technical specifications such as the exact model of the camera, resolution, and camera exposure to increase the realism of the experiment.
3,Evaluation Metrics
How does the total Blob area reflect the sensitivity of the algorithm to changes in texture and lighting conditions?
4,Experimental Results
How are the performance differences of different Blod detection algorithms (Adaptive Thresholding, LoG, DoG, and DoH) under different lighting conditions (IR irradiation vs. natural light irradiation) represented by experimental data?
The evaluation of algorithms focuses on finding the optimal balance between sensitivity and accuracy under different light and algorithm configurations. How is the “optimal balance” assessed and defined? How to choose the best threshold setting to prevent too many false positives or noise when dealing with different types of pollutants?
Do different color temperatures and light intensities affect the stability of Blod detection methods?
5,Conclusions
The conclusion section mentions that future work will focus on algorithmic robustness and real-time processing improvements, but this section lacks a connection to the previous section. Is it possible to further explain in the conclusion how these future improvements specifically respond to the challenges in the current research?
Reviewer 3 Report
Comments and Suggestions for Authors
-
The abstract is too brief. It is recommended that the authors expand it and include the key data results achieved in this study to better demonstrate its value.
-
I suggest the authors clearly specify which type or model of automotive parts was tested in the experiment, so that readers can better understand the context of the study.
-
In the description of the preliminary work, the authors should consider providing details about the imaging equipment used, including its specifications and the exact shooting location. If a professional imaging platform was used, it should also be described.
-
The data analysis in the current tables is limited. The authors could supplement the manuscript with a performance comparison between the proposed method and several classical threshold-based segmentation methods.
-
A discussion section should be added to analyze the limitations of this study and outline possible directions for future research.
Round 2
Reviewer 3 Report
Comments and Suggestions for Authors
The author addressed all my suggestions.